# Hierarchical MAB Framework for Energy-Aware Beam Training for Near-Field Communications

**DOI:** 10.3390/s26010060

**Published:** 2025-12-21

**Authors:** Yunxing Xiang, Yi Yan, Yunchao Song, Jing Gao, Xiaohui You, Jun Wang, Huibin Liang, Yixin Jiang

**Affiliations:** 1Supply Service Management Center, State Grid Jiangxi Electric Power Co., Ltd., Nanchang 330001, China; px_xiangyx@jx.sgcc.com.cn (Y.X.); dky_yanyi@jx.sgcc.com.cn (Y.Y.); gfzx_gaojing@jx.sgcc.com.cn (J.G.); gx_youxh@jx.sgcc.com.cn (X.Y.); dky_wangjun1@jx.sgcc.com.cn (J.W.); 2College of Electronic and Optical Engineering, Nanjing University of Posts and Telecommuncations, Nanjing 210023, China; 2021020305@njupt.edu.cn (H.L.); 1023020617@njupt.edu.cn (Y.J.)

**Keywords:** XL-MIMO, beam training, user scheduling, MAB, energy efficiency, KL-UCB

## Abstract

For XL-MIMO multi-user frequency division duplex systems, this paper proposes a near-field beam training scheme using a two-phase combinatorial multi-armed bandit (MAB) framework. This scheme leverages the MAB framework, integrating energy-aware user scheduling and hierarchical beam training to balance communication quality and device battery level, thereby effectively enhancing system energy efficiency and extending the device’s lifespan. Specifically, in the first phase, we account for user battery levels by designing an energy-aware upper confidence bound (UCB) algorithm for user scheduling. This algorithm effectively balances exploration and exploitation, prioritizing users with higher achievable rates and sufficient battery level. In the second phase, based on the scheduled users, two UCB algorithms are employed for beam training. In the first layer, discrete Fourier transform codebook-based beam scanning is utilized, and a UCB algorithm is applied to initially acquire angle information for scheduled users. In the second layer, based on the obtained angle information, a candidate set of polar-domain codewords is constructed. Another UCB algorithm is then employed to select the optimal polar-domain codewords. The effectiveness of our scheme is confirmed by simulations, demonstrating notable achievable rate gains for multi-user communications.

## 1. Introduction

The expanded spatial degrees of freedom in Extremely Large-Scale Multiple-Input Multiple-Output (XL-MIMO) facilitate simultaneous improvements in frequency utilization, channel capacity, and transmission delays, paving the way for advanced wireless architectures [1,2,3]. However, the substantial increase in array scale also presents new theoretical and practical challenges. In particular, as the propagation scenario moves to the near-field scenario, the traditional scheme in far-field model becomes inadequate, invalidating the planar wave approximation and necessitating the incorporation of wavefront curvature effects. This shift necessitates adaptations in critical processes such as channel modeling, beam training, and user scheduling [4,5,6].

To tackle the excessive complexity and overhead associated with near-field beam training, researchers have recently proposed various schemes aimed at reducing resource consumption while maintaining accuracy. For example, by modeling inter-user relationships with a Graph Neural Network, the three-phase method proposed in [7] for multi-user XL-MIMO systems achieves higher beam estimation accuracy and substantially lower pilot overhead. Another widely studied approach adopts a two-stage hierarchical search framework, such as using a central sub-array for coarse angular search to determine the general user direction, followed by refined estimation in the polar domain jointly optimizing angle and distance [8]. Building on this, Lu et al. proposeded a multi-resolution codebook design [9], enabling two-dimensional hierarchical near-field beam training through wide-beam coarse search followed by progressive refinement, thereby avoiding the high overhead of an exhaustive search. A substantial reduction in training overhead was achieved in [10] through an auxiliary codebook based on spatial chirp beams. This method analyzes beam patterns in the slant-intercept domain to capitalize on the coupling relationship between the angle and distance. More recently, a sparse discrete Fourier transform (DFT) codebook-based three-stage beam training method has been introduced: first performing sparse scanning in the angular subspace to select candidate angles, then activating the central sub-array to resolve angle ambiguity, and finally searching for the optimal distance within the estimated angle using a polar-domain codebook [11]. These studies have laid an important foundation for transitioning near-field XL-MIMO systems from theory to practice. However, most existing studies focus primarily on the beam training process within a single coherence block interval and fail to fully exploit the correlation of channel characteristics across different coherence periods. In contrast, statistical channel state information (CSI) often remains relatively stable over longer time scales [12], offering new opportunities for designing low-overhead beam training schemes.

Meanwhile, within the context of near-field communications for XL-MIMO systems, the multi-user scheduling problem also requires re-evaluation. A variety of specialized strategies have been introduced for XL-MIMO systems, seeking to optimize system performance through different technical avenues. By integrating coalition-search-based scheduling with optimal power allocation, the work in [13] addresses joint user scheduling and power allocation with a quality-of-service awareness. This approach improves user accommodation in dense networks while guaranteeing minimum user rates. Building upon this, to better adapt to channel uncertainties in practical systems, the authors [14] considered the impact of imperfect CSI, employed statistical CSI (S-CSI) for user scheduling, and jointly designed precoding to improve overall spectral efficiency and reduce training overhead. Recently, user scheduling mechanisms have begun to integrate deeply with specific application scenarios. A work focused on maritime unmanned aerial vehicle (UAV)-assisted communication, deriving orthogonal positions in the near-field region and jointly optimizing UAV altitude and user scheduling to effectively reduce channel interference and improve the system sum rate [15]. Beyond channel-state-based methods, geometry-based scheduling strategies have garnered attention due to their effectiveness in reducing complexity. The authors [16] proposed a distance-based scheduling mechanism that classifies users based on their equivalent distance relative to the antenna array, significantly reducing computational complexity while mitigating inter-user interference. A similar interference management approach has been extended to multi-cell systems. The authors [17] proposed a novel user scheduling strategy for multi-cell near-field MIMO systems, clustering users from the perspectives of both inter-user interference and inter-cell interference, thereby achieving effective management of system interference and a reduction in scheduling complexity. However, these existing user scheduling schemes generally overlook the energy status of user equipment and fail to incorporate terminal battery level constraints into the scheduling decision process [18]. This limitation becomes particularly critical in practical energy-constrained communication scenarios. It may lead to frequently scheduling users with an insufficient battery level, thereby accelerating their energy depletion, shortening device lifespan, and potentially causing connection interruptions, ultimately compromising overall system energy efficiency and user experience.

Existing beam training schemes for near-field XL-MIMO face several limitations. Symmetric beam training methods suffer from beam absence at certain angular regions [19]. Array partition techniques can mitigate this but require extra calibration beams, increasing training overhead [20]. For user scheduling, many approaches lack adaptability or key practical considerations. Supervised learning models [21] cannot handle dynamic environments, while deep reinforcement learning methods [22] often ignore terminal energy constraints. Even near-field scheduling schemes using statistical CSI [23] focus solely on received energy, lacking a joint energy-distance awareness.

In this context, the focus of this paper is the investigation of integrated multi-user beam training and scheduling strategies tailored for XL-MIMO. We propose a two-phase combinatorial multi-armed bandit (MAB) beam training framework. In the first phase, we propose an energy-aware upper confidence bound (UCB) algorithm to address the user scheduling problem. This algorithm utilizes the average achievable rate as the reward, jointly considering user channel quality and battery level. The energy-aware UCB (EA-UCB) value consists of three components: the historical average achievable rate, an uncertainty term, and a penalty term related to the remaining battery level. Maximizing the EA-UCB value selects users that are most likely to deliver superior rates within each coherence block, thereby facilitating efficient scheduling. In the first layer of the second phase, we employ a DFT codebook-based beam scanning, and apply a Kullback–Leibler UCB (KL-UCB) algorithm to obtain angle information for the scheduled users. Subsequently, with the acquired angle information, the second layer proceeds to construct a corresponding candidate set of polar-domain codewords. Again, using a KL-UCB algorithm with beam gain as the reward, we select the polar-domain codeword that maximizes the average beam gain for the scheduled users. The contributions are concluded as follows:An energy-aware user scheduling algorithm that addresses the common oversight of terminal energy efficiency in existing mechanisms is developed. By incorporating the residual energy status of devices into UCB strategy as a weighting factor, our approach dynamically prioritizes users with favorable channel conditions and sufficient energy. This strategy not only balances exploration and exploitation, but also significantly enhances overall network energy efficiency, extends the operational lifespan of battery-constrained devices, and improves service fairness in energy-heterogeneous scenarios.This paper formulates an integrated framework that co-optimizes user scheduling and near-field beam training (US-NFBT), overcoming the limitations of conventional decoupled designs. The proposed method employs a two-phase combinatorial (C-MAB) model to first carry out energy-aware user selection, followed by a dual-layer beam training mechanism. This hierarchical process begins with coarse-grained angle estimation and advances to fine-grained polar-domain beam training, leading to a substantial reduction in training overhead. As a result, the proposed scheme effectively mitigates the resource consumption problem introduced by the additional ranging requirement inherent to near-field communications.A codebook-based non-reciprocal beam training scheme is designed. By efficiently acquiring downlink channel information using a triple UCB strategy with integrated device battery awareness, a practical and energy-efficient multi-user near-field communication solution is offered. This approach not only improves beam training performance, but also extends device battery life, thereby facilitating the real-world deployment of frequency division duplex (FDD) XL-MIMO systems.

The proposed framework directly addresses these limitations by integrating several synergistic innovations. A dual-layer KL-UCB beam training structure progressively narrows the search space, effectively pruning redundant beams to reduce overhead. Simultaneously, a novel energy-aware UCB scheduler incorporates a distance–energy penalty term, dynamically balancing the communication load to prolong network lifetime. Furthermore, by leveraging the temporal stability of statistical CSI instead of instantaneous CSI, the framework minimizes the need for frequent and costly channel estimation. Together, these design choices form a cohesive and practical solution for low-overhead, energy-efficient multi-user communications in the near-field.

## 2. System Model

### 2.1. Communication Scenario and Channel Model

As shown in Figure 1, this work studies an FDD XL-MIMO downlink system. The base station (BS) is equipped with a uniform linear array (ULA). The ULA has *N* antennas and $M_{\mathrm{RF}}$ RF chains. A ULA aligned with the y-axis is considered, where the coordinate of the *n*-th antenna element (for n=1,⋯,N) is $\left(0,\delta_{n}d\right)$, with antenna spacing d=λ/2 defined by the carrier wavelength λ. The BS communicates with *K* users, all located within the array near-field, i.e., their distances from the BS are less than the Rayleigh distance [9]. Each user has one antenna. The channel model adopts the spherical wave assumption. Consider a multiple scattering clusters channel, during the *t*-th coherence block, the channel vector $h_{t}^{k}$ from the BS to user *k* is [24](1)$$h_{t}^{k}=a_{k}\sum_{s=1}^{S}\sum_{l=1}^{L_{s}}\frac{\lambda g_{l}^{s}e^{-j\frac{2\pi }{\lambda }\left(r_{l}^{s}+\mu_{l}^{s}\right)+jw_{l}^{s}}}{{\left(4\pi \right)}^{3/2}r_{l}^{s}\mu_{l}^{s}}b(\theta_{l}^{s},r_{l}^{s}),$$
where $a_{k}=\sqrt{\frac{N}{\sum_{s=1}^{S}L_{s}}}$ is a normalization constant, which ensures per-user power normalization, accounting for the total number of paths across all clusters associated with user *k*, *S* is the number of clusters, $L_{s}$ is the number of paths within the *s*-th cluster, $\mu_{l}^{s}$ is the distance between the reference point and the user, and $w_{l}^{s}$ is a phase shift, characterized as a variable followed by a uniform distribution across the interval [−π,π). $g_{l}^{s}∼\mathrm{CN}\left(0,\sigma_{l}^{s,2}\right)$ is the *l*-th path gain in the *s*-th cluster, where $\sigma_{l}^{s}$ is uniformly distributed in [0,1]. The channel model in Equation (1) is adapted from Equations (33)–(34) in [24], with modifications to accommodate single-antenna users in our system. Unlike [24], where users employ multiple antennas, our model eliminates the user-side steering vector, thus simplifying the expression.

The spherical wave array steering vector in the near-field, denoted as $b(\theta_{l}^{s},r_{l}^{s})$, is(2)$$b(\theta_{l}^{s},r_{l}^{s})=\sqrt{\frac{1}{N}}{\left[e^{-j\frac{2\pi }{\lambda }\left(r_{l}^{s,(0)}-r_{l}^{s}\right)},\ldots ,e^{-j\frac{2\pi }{\lambda }\left(r_{l}^{s,(N-1)}-r_{l}^{s}\right)}\right]}^{T},$$
where $r_{l}^{s,(n)}=\sqrt{{\left(r_{l}^{s}\sqrt{1-{\left(\sin\theta_{l}^{s}\right)}^{2}}-0\right)}^{2}+{\left(r_{l}^{s}\sin\theta_{l}^{s}-nd\right)}^{2}}=\sqrt{{\left(r_{l}^{s}\right)}^{2}+{\left(nd\right)}^{2}-2ndr_{l}^{s}\sin\theta_{l}^{s}},$ gives the propagation path length from the *n*-th BS antenna to the scatterer in the *s*-th cluster, while $r_{l}^{s}$ and $\theta_{l}^{s}$ represent the *l*-th path’s distance and angle parameters, respectively. This differs from the cosine term in ([24], Equation (2)), which describes the distance from the user’s receiving antenna to the cluster.

The received signal $y_{t}=\left[y_{1,t},\ldots ,y_{K,t}\right]\in C^{K\times 1}$ in the *t*-th coherence block is(3)$$y_{t}=H_{t}^{H}W_{t}s_{t}+n_{t},$$
where $H_{t}=\left[h_{t}^{1},\ldots ,h_{t}^{K}\right]$ is the channel matrix, $W_{t}=D_{t}V_{t}=\left[w_{1,t},\ldots ,w_{K,t}\right]\in C^{N\times K}$ is the beamforming matrix, $D_{t}\in C^{N\times |G_{t}|}$ is the codebook-based analog beamforming matrix designed through beam training, $V_{t}$ is a digital precoding matrix, which can be designed based on the zero-forcing criterion [25], $s_{t}={\left[s_{1},\ldots ,s_{K}\right]}^{T}\in C^{K\times 1}$ is the signal vector transmitted from the base station obeying $E(s_{t}s_{t}^{H})=I$, and $n_{t}∼\mathrm{CN}\left(0,\sigma^{2}I\right)$ is the AWGN.

### 2.2. Energy Model

The system-wide power consumption comprises two components: the transmit power consumption and the fixed circuit power consumption. Specifically, under the considered communication scenario and channel model for user *k*, the system-wide power consumption follows the model in [26](4)$$\xi_{k}^{\mathrm{total}}=\gamma {∥w_{k,t}∥}_{2}^{2}+\xi_{\mathrm{system}},$$
where 1/γ denotes the power amplifier efficiency, $w_{k,t}$ represents the beamforming vector for user *k* at the *t*-th coherence block, and $\xi_{\mathrm{system}}$ denotes the system circuit power.

The user scheduling decision is represented by a binary variable $G_{k,t}$ [27](5)$$G_{k,t}=\left\{\begin{array}{cc} 1, & \mathrm{if}\;\mathrm{user}\;k\;\mathrm{is}\;\mathrm{selected};\\ 0, & \mathrm{otherwise}. \end{array}\right.$$The subset of scheduled users is denoted by $G_{t}$, where $G_{t}=\left\{G_{\phi_{1,t},t},\ldots ,G_{\phi_{j,t},t}\right\}$, $\phi_{j,t}\in \left\{1,2,\cdots ,K\right\}$, $|G_{t}|\leq M$ and *M* is the maximum schedulable user number.

The residual energy of a scheduled user *k* is updated according to [18](6)$$\Psi_{k,t}=\Psi_{k,t-1}-\frac{\xi_{k}^{\mathrm{total}}L_{\mathrm{data}}}{W_{\mathrm{bw}}E_{k,t}},$$
where $L_{\mathrm{data}}$ denotes the number of bits of the requested data obtained by scheduled user *k* per coherence block, $W_{\mathrm{bw}}$ denotes the bandwidth, and $E_{k,t}$ represents the effective achievable rate (EAR) [28], which is(7)$$E_{k,t}=G_{k,t}\left(\frac{T_{\mathrm{total}}-T_{B,t}}{T_{\mathrm{total}}}\right)\log_{2}\left(1+\frac{P_{\mathrm{signal}}}{P_{\mathrm{interference}}+\sigma^{2}}\right),$$
where $T_{B,t}$ and $T_{\mathrm{total}}$ denote the beam training overhead and the total symbol count per coherence block, respectively. $P_{\mathrm{signal}}={\left|(h_{t}^{k}{)}^{H}w_{k,t}\right|}^{2}$ represents the received signal power, and $P_{\mathrm{interference}}=\sum_{n\neq k}{\left|{\left(h_{t}^{k}\right)}^{H}w_{n,t}\right|}^{2}$ denotes the aggregate interference power from other users. Then, the total EAR is $E_{t}=\sum_{k=1}^{K}E_{k,t}$. The term $\frac{\xi_{k}^{\mathrm{total}}L_{\mathrm{data}}}{W_{\mathrm{bw}}E_{k,t}}$ models the energy consumption per $L_{\mathrm{data}}$ bit data transmitted. Note that if $\Psi_{k,t}<\Psi_{\mathrm{th}}$, user *k* will not be scheduled in subsequent coherence blocks, where $\Psi_{\mathrm{th}}$ is a predefined energy threshold. For the subsequent analysis, we set ${\tilde{E}}_{k,t}=\log_{2}\left(1+\frac{P_{\mathrm{signal}}}{P_{\mathrm{interference}}+\sigma^{2}}\right)$.

The energy consumption model employed in this work follows established formulations adopted in the literature [18,29,30]. As described by Equation (4), the system-wide power consumption per user incorporates a transmit power term $\gamma {∥w_{k,t}∥}_{2}^{2}$, proportional to the beamforming gain and directly modeling the dominant consumption of the radio frequency power amplifier [31], alongside a constant circuit power component $\xi_{\mathrm{system}}$ accounting for the static power draw of essential baseband and frequency synthesis circuits. This physical consistency is maintained in the user-side residual energy update of Equation (6), where the total energy consumed for transmitting a fixed data volume $L_{\mathrm{data}}$ depends on the required transmission time $\frac{L_{\mathrm{data}}}{W_{\mathrm{bw}}E_{k,t}}$. Since this transmission time is inversely proportional to the effective achievable rate $E_{k,t}$, the model captures a key practical energy efficiency principle: poor channel conditions (lower rates) necessitate longer transmission times and thus higher energy consumption, accelerating battery drain, whereas favorable conditions enable more energy-efficient data transfer.

The model is both physically grounded and adaptable to various terminal types through parameter adjustment. For battery-limited IoT devices [29], it accurately reflects their typical energy profile, enabling scheduling algorithms to extend network lifetime by managing the energy expenditure of nodes. For more complex devices like smartphones, the circuit power parameter $\xi_{\mathrm{system}}$ can be calibrated to represent their higher active power, allowing the scheduler to balance consumption across diverse users. Overall, this formulation uses a concise set of meaningful parameters to facilitate energy-aware scheduling across a wide range of practical wireless scenarios.

### 2.3. Problem Formulation

Most conventional near-field beam training methods overlook user energy disparities and lack integrated scheduling. To address this, we introduce a scheduling strategy that selects users for service during each coherence block. This selection jointly considers two factors: channel quality and residual energy levels. The aim is to maximize the total EAR throughout the S-CSI invariance interval.

To this end, it is imperative to have an effective user scheduling strategy coupled with a well-designed beamforming matrix. To mitigate CSI acquisition overhead, the adoption of codebook-based analog beamforming is widespread in terrestrial systems [32]. Typically, such an analog beamforming matrix is constructed from a predefined codebook. The polar-domain codebook, as a predefined codebook choice, offers a distinct advantage over its DFT-based counterpart for near-field model [33]. In the polar-domain codebook $D^{n}$, the *i*-th codeword $b\left(\theta_{i},r_{i}\right)$ is formulated using the near-field steering vector.

Assuming S-CSI remains constant over *T* coherence blocks, the problem is formulated as(8)P:maxWt,Gk,t∑t=1T∑k=1KEk,t(8a)s.t.∑kGk,t≤M(8b)Ψk,t≥Ψth,∀k∈1,2,⋯,K(8c)dk,t∈Dn,∀k∈1,2,⋯,K.To reduce computational complexity and training overhead, we propose a two-phase scheme for jointly perform user scheduling and optimal codeword selection. In the first phase, a user subset is selected through scheduling. In the second phase, the codebook-based analog beamforming matrix $D_{t}$ is optimized for this subset. We formulate this joint design problem and introduce a MAB-based beam training scheme as our solution, detailed in the subsequent section.

## 3. Two-Phase Near-Field Beam Training Scheme with User Scheduling

A reinforcement learning framework, called the MAB, is designed for sequential decision-making, maximizing the cumulative reward through an explore–exploit trade-off. This framework has been successfully applied to various challenges in wireless communications, such as user scheduling and/or beam training. Instances include using deep contextual bandits for near-optimal mmWave beam selection [34], MAB-based pilot allocation for efficient beam alignment [35], and contextual MAB for dynamic user scheduling in massive MIMO with fairness considerations [36]. A recent advancement employs a hierarchical MAB with a dual-UCB strategy for joint beamforming and user scheduling in low earth orbit satellite networks, enabling pilot-free acquisition of S-CSI from historical data to greatly enhance net spectral efficiency [23]. The aforementioned works demonstrate the potential of MAB-based solutions in addressing key challenges within the field of wireless communications.

Inspired by these developments, we introduce a C-MAB-based beam training scheme that operates in two phases. Our scheme, depicted in Figure 2, operates within each coherence block through a sequence of four steps: user scheduling, beam training, digital precoding, and data transmission. Specifically, in the first phase, the scheduling strategy is determined by both the channel quality and the residual energy levels of the users. The second phase subsequently performs angular scanning followed by polar-domain scanning. This decomposition breaks down the original optimization problem P into three coupled subproblems: user scheduling, angular scanning, and polar-domain scanning.

### 3.1. EA-UCB-Based User Scheduling

In recent years, UCB and its variants have found widespread applicability in wireless network optimization. Examples include correlation-aware link rate selection (related to Min-UCB) [37], outage-based meta-scheduling for downlink systems [38], and multi-agent task offloading algorithms in edge computing [39]. Motivated by these advances, in the first phase, we propose an EA-UCB algorithm for user scheduling. Our EA-UCB algorithm dynamically selects an optimal user subset in each coherence block, achieving joint optimization of communication performance and energy consumption. Unlike conventional UCB methods that focus solely on channel state or throughput, the proposed EA-UCB incorporates multiple dimensions, including instantaneous achievable rate, historical scheduling experience, and residual battery level into a unified decision-making framework.

For each user *k*, the scheduling merit metric (EA-UCB value) [40] is defined as follows(9)$${\mathrm{EU}}_{k}\left(t\right)=\underset{\left(a\right)}{\underset{︸}{{\bar{E}}_{k,t}+\sqrt{\frac{B\ln t}{C_{k,t}}}}}-\underset{\left(b\right)}{\underset{︸}{\frac{r_{k}^{\prime }}{\Psi_{k,t}}}},$$
where ${\bar{E}}_{k,t}$ represents the historical average achievable rate (excluding the overhead term) of user *k* up to the *t*-th coherence block, $C_{k,t}$ gives the total scheduling count for user *k* up to the *t*-th coherence block, *B* is an exploration coefficient that controls the degree of exploration, and $r_{k}^{\prime }$ indicates a normalized parameter based on the distance between user *k* to the BS.

In Equation (Equation 9), component (a) embodies the core principle of the classical UCB algorithm, which seeks to balance exploitation of known information and exploration of new alternatives during decision-making. Here, ${\bar{E}}_{k,t}$ favors users with a historically good performance, representing exploitation. The second term, $\sqrt{\frac{B\ln t}{C_{k,t}}}$, serves as an exploration bonus, where $C_{k,t}$ denotes the number of times user *k* has been scheduled so far. By being inversely proportional to the scheduling count, this term encourages sampling less-scheduled users, thereby helping to discover potentially superior options. Collectively, component (a) acts as an optimistic estimate of user performance, promoting both the selection of top-performing users and sufficient exploration of less frequently scheduled ones, which enables the method to effectively converge to an optimal user combination in dynamic propagation conditions.

This subsection formally defines the distance–energy ratio as a key metric, denoted as component (b): $\frac{r_{k}^{\prime }}{\Psi_{k,t}}$. This metric integrates energy availability and distance-related paths into a proportional penalty term. A higher value of $\frac{r_{k}^{\prime }}{\Psi_{k,t}}$ indicates that a user is either severely energy-constrained or located farther from the base station, resulting in a lower scheduling priority. This strategy enhances energy efficiency and extends network longevity without sacrificing system throughput. By jointly considering near-field path loss and energy consumption, the approach reduces the priority given to users with poor channel conditions or insufficient energy, leading to more efficient resource allocation.

Then, the user scheduling problem is formulated as follows(10)P1:argmaxGtE¯k,t+BlntCk,t−rk′Ψk,t(10a)s.t.Gt≤M,
where the number of users scheduled at the *t*-th coherence is $M_{t}=\left|G_{t}\right|$. Algorithm 1 details the proposed EA-UCB-based user scheduling strategy. Given that $R_{k}=E\left(\left(h_{t}^{k}\right){\left(h_{t}^{k}\right)}^{H}\right)$, $E\left(P_{\mathrm{signal}}\right)\leq \mathrm{Trace}\left(R_{k}\right)\leq N$ and $E\left(P_{\mathrm{interference}}\right)\geq 0$, it follows that $E\left({\tilde{E}}_{k,t}\right)\leq \log_{2}\left(1+\frac{N}{\sigma^{2}}\right)$. Thus, we set ${\bar{E}}_{k,1}=\log_{2}\left(1+\frac{N}{\sigma^{2}}\right)$ for all *k*.

**Remark** **1.**
*Users whose EA-UCB values are negative or whose residual energy falls below the threshold should be excluded to avoid inefficient scheduling of energy-depleted devices.*


With pre-screening conditions that mandate non-negative EA-UCB values and residual energy above a threshold, the algorithm efficiently disqualifies users not meeting basic criteria, thus shrinking the solution space and boosting real-time performance. The embedding of energy state into the scheduling value function allows the EA-UCB method to achieve higher system energy efficiency and longer device operational life, without greatly increasing computational complexity.
**Algorithm 1** EA-UCB-based user scheduling**Input:** $\Psi_{\mathrm{th}}$, *M*, $r_{k}$, $\Psi_{k,1}$1:Initialize ${\bar{E}}_{k,1}=\log_{2}\left(1+\frac{N}{\sigma^{2}}\right)$ and $C_{k,1}=1$ for all *k*2:Initialize $G_{t}\leftarrow Ø$, $V_{t}\leftarrow Ø$ {Initialize valid user set}3:**for** t=2 **to** *T* **do**4:   **for** each user *k* **do**5:       **if** $\Psi_{k,t}\geq \Psi_{\mathrm{th}}$ **then**6:           Compute ${\mathrm{EU}}_{k}\left(t\right)={\bar{E}}_{k,t}+\sqrt{\frac{B\ln t}{C_{k,t}}}-\frac{r_{k}^{\prime }}{\Psi_{k,t}}$7:           **if** ${\mathrm{EU}}_{k}\left(t\right)>0$ **then**8:               Add *k* to $V_{t}$9:           **end if**10:      **else**11:           ${\mathrm{EU}}_{k}\left(t\right)\leftarrow -\infty$12:      **end if**13:   **end for**14:   $M_{t}=\min\left(M,\left|V_{t}\right|\right)$ Determine number of users to schedule15:   Select top $M_{t}$ users from $V_{t}$ with highest ${\mathrm{EU}}_{k}\left(t\right)$ to form $G_{t}$16:   Run Algorithm 2 to get the selected polar-domain codewords17:   Compute achievable rate (excluding the overhead term) ${\tilde{E}}_{k,t}$ for each scheduled user18:   Update for each scheduled user $k\in G_{t}$:19:         $C_{k,t+1}\leftarrow C_{k,t}+1$20:         ${\bar{E}}_{k,t+1}\leftarrow \frac{{\bar{E}}_{k,t}C_{k,t}+{\tilde{E}}_{k,t}}{C_{k,t+1}}$21:         $\Psi_{k,t+1}\leftarrow \Psi_{k,t}-\frac{\xi_{k}^{\mathrm{total}}L_{\mathrm{data}}}{W_{\mathrm{bw}}E_{k,t}}$22:**end for****Output:**  $G_{t}$
**Algorithm 2** Two-layer MAB-Based Beam Training**Input:** $D^{f}$, $D^{n}$, $G_{t}$1:Initialize $C_{i,t}^{f}=0$ and $C_{i,t}^{n}=0$2:Initialize the first and second layers base arms3:**for **t=2 **to** *T* **do**4:   **Layer 1: Angular Scanning (DFT Codebook)**5:   Compute ${\mathrm{KU}}_{i}^{f}\left(t\right)$ via the bisection method6:   Select $M^{f}$ codewords to constitute $A_{t}^{f}$ (**Remark 2**)7:   **Layer 2: Polar-domain scanning (constructed polar-domain codebook)**8:   Construct candidate angle set $\bar{\Theta }$ from $A_{t}^{f}$9:   Construct candidate codebook ${\bar{D}}^{n}$ from $\bar{\Theta }$10:   Compute ${\mathrm{KU}}_{i}^{n}\left(t\right)$ via the bisection method11:   Select $M^{n}$ codewords to constitute $A_{t}^{n}$ (similar to **Remark 2**)12:   Update $C_{i,t+1}^{f}$, $C_{j,t+1}^{n}$, ${\bar{E}}_{i,t+1}^{f}$, ${\bar{E}}_{j,t+1}^{n}$ using received signals13:**end for****Output:**  $A_{t}^{f}$, $A_{t}^{n}$

### 3.2. MAB-Based Angular Scanning

We note that for the DFT codebook, the near-field channel exhibits a comparable energy distribution within angular ranges where the energy spreads towards the neighboring angles. This allows the DFT codebook to effectively perform initial angular-domain sweeping in the near-field as a practical starting point. Figure 3 shows the beam gain of the DFT codebook under far-field and near-field paths, where $\Delta_{\theta }=\cos\left(\theta_{c}\right)-\cos\left(\theta_{p}\right)$, $\theta_{c}$ is the corresponding angle of the codeword in the DFT codebook, and $\theta_{p}$ is the angle of the path. As illustrated in Figure 3, angular domain sweeping with the DFT codebook yields an exact angle for users in the far-field; for near-field users, it provides a broad angular range, within which subsequent polar domain sweeping using the polar-domain codebook can precisely locate the user.

In the first layer of the second phase, this paper adopts a DFT codebook $D^{f}$ for angular scanning. The goal is to determine some codewords in the codebook that maximize the sum performance gain at the scheduled users, thereby providing a candidate degree set for the subsequent polar-domain codebook construction.

Let $M^{f}$ denote the training overhead, and let $A_{t}^{f}$ represent the collection of indices for the selected DFT codewords within the *t*-th block. Denote $d_{i}^{f},\;i=1,\cdots ,N$ as the *i*-th DFT codeword and $D^{f}=\left\{d_{1}^{f},\cdots ,d_{N}^{f}\right\}$. The optimization problem can be formulated as selecting a codeword set $A_{t}^{f}$ to maximize the beam gain for the scheduled users, expressed as(11)P2:maxAtf∑t=1T∑k∈Gt∑i∈AtfhtkHdif(11a)s.t.dif∈Df.

Within information theory and statistical learning, divergence functions serve as a fundamental tool for measuring differences between probability distributions. A prominent example is the asymmetric Kullback–Leibler (KL) divergence [41], commonly employed to evaluate the difference between a true distribution *Q* and an approximating distribution $\tilde{Q}$, with its discrete form given by $f_{\mathrm{KL}}\left(Q|\right|\tilde{Q})=\int_{-\infty }^{\infty }Q\left(x\right)\log\frac{Q(x)}{\tilde{Q}\left(x\right)}dx$. Theoretically, KL-UCB leverages the KL divergence to construct tighter confidence bounds than the standard Hoeffding inequality-based UCB. This enables a more precise characterization of tail behavior in probability distributions, leading to superior asymptotic regret performance. For arm *i*, we define the UCB value as(12)$${\mathrm{KU}}_{i}\left(t\right)=\max\left\{\tilde{\mu }:C_{i,t}f_{\mathrm{KL}}\left({\bar{\mu }}_{i}\left|\right|\tilde{\mu }\right)\leq f_{\mathrm{expl}}\left(t\right)\right\},$$
where $f_{\mathrm{expl}}\left(t\right)=\log\left(t\right)+c\cdot \log\left(\log\left(t\right)\right)$ is an exploration term, with *c* as a tuning coefficient, ${\bar{\mu }}_{i}$ means the sample average reward of arm *i*, and $\tilde{\mu }$ is a candidate value for the theoretical expected reward to be solved for.

Conventional optimization methods based on channel covariance matrices (CCMs) are often ineffective in practice due to the lack of S-CSI. To address this challenge, this paper employs a MAB framework and adopts a KL-UCB strategy for efficient problem-solving. The CCM, a key component of S-CSI, varies at a much slower rate than instantaneous CSI (I-CSI) [12]. Thus, it is considered approximately constant over multiple channel coherence intervals. For each user, its corresponding CCM $R_{k}$ remains unchanged over an extended period *T*.

We formulate the aforementioned optimization problem $P_{2}$ as a C-MAB problem oriented towards the scheduled users, and employ the KL-UCB algorithm to solve it. In this model, each codeword $d_{i}^{f}$ in the DFT codebook $D^{f}$ is treated as a base arm, while the set of selected DFT codewords in each training round forms a super arm, denoted as $A_{t}^{f}$. In the *t*-th coherence block, for any scheduled user *k*, the reward associated with the *i*-th base arm is set to user *k*’s detected signal power when the *i*-th codeword is used. This reward is expressed as $E_{i,t}^{f}={\left|y_{k,i,t}^{f}\right|}^{2}={\left|{\left(h_{t}^{k}\right)}^{H}d_{i}^{f}+n_{k,t}\right|}^{2}$.

We now give lemmas that are useful to analyze the distribution form of ${\left|y_{k,i,t}^{f}\right|}^{2}$ and the corresponding KL divergence form for this distribution.

**Lemma** **1.**
*For a random variable A∼CN(0,κ), the squared modulus ${\left|A\right|}^{2}$ follows an exponential distribution, i.e., ${\left|A\right|}^{2}∼\mathrm{Exp}\left(\frac{1}{\kappa }\right)$.*


**Proof.** See Appendix A. □

Therefore, ${\left|y_{k,i,t}^{f}\right|}^{2}$ follows an exponential distribution with parameter $\frac{1}{{\left(d_{i}^{f}\right)}^{H}R_{k}\left(d_{i}^{f}\right)+\sigma^{2}}$. Based on Lemma 1, we can obtain the following lemma.

**Lemma** **2.**
*The derivation of the KL divergence is*

(13)
$$f_{\mathrm{KL}}\left(q|\right|\tilde{q})=\log\left(\frac{\lambda_{q}}{\lambda_{\tilde{q}}}\right)+\frac{\lambda_{\tilde{q}}}{\lambda_{q}}-1,$$

*where $q\left(x\right)=\lambda_{q}e^{-\lambda_{q}x}$ and $\tilde{q}\left(x\right)=\lambda_{\tilde{q}}e^{-\lambda_{\tilde{q}}x}$ are exponential with parameter $\lambda_{q}$ and $\lambda_{\tilde{q}}$, respectively.*


**Proof.** See Appendix B. □

**Theorem** **1.**
*The first layer of the proposed beam training method achieves an expected regret of O(lnt).*


**Proof.** The proof follows a similar procedure to that in Section 3 of [41], and is therefore omitted here for brevity. □

Theorem 1 guarantees that, under the condition that CCM remains constant, the regret of the first layer grows logarithmically. This enables the algorithm to asymptotically converge to the optimal codewords in the DFT codebook, thereby maximizing the expected beamforming gain.

**Remark** **2.**
*We restrict the number of codewords chosen per scheduled user k to $A_{t,k}^{f}=\left\{j:\frac{1}{{KU}_{j}}\left(t\right)\geq \rho \cdot \max_{i}\;\frac{1}{{KU}_{i}\left(t\right)}\right\}$ following the computation of the KL-UCB values, such that the training overhead can be reduced. Consequently, this leads to $A_{t}^{f}={⋃}_{k\in G_{t}}A_{t,k}^{f}$ and $M^{f}=\left|A_{t}^{f}\right|$.*


In the proposed scheme, the system progressively learns the average beam gain of each DFT codeword. Specifically, the average beam gain of user *k* at the *i*-th codeword $d_{i}^{f}$ is denoted as $\varepsilon_{k,i}^{f}=E\left({\left|{\left(h_{t}^{k}\right)}^{H}d_{i}^{f}\right|}^{2}\right)={\left(d_{i}^{f}\right)}^{H}R_{k}d_{i}^{f}$. Note that codewords with higher average beam gains are more likely to provide high beamforming gains for the users across multiple coherence intervals. By prioritizing the selection of these high-gain codewords and reducing the exploration of low-gain ones, system performance is therefore guaranteed.

### 3.3. MAB-Based Polar-Domain Scanning

Building upon the angle information acquired in the previous Section 3.2, we filter candidate polar-domain codewords for polar-domain scanning, and then a C-MAB-based polar-domain scanning strategy is proposed.

In the second layer of the second phase, we leverage the angular scanning results from the first layer to extract angle information for each scheduled user and construct a refined set of angle indices. Based on this, we obtain the corresponding candidate set of polar-domain codeword angles $\bar{\Theta }=\left\{\theta_{i}|i\in A_{t}^{f}\right\}$, where $\theta_{i}=\frac{2i-N-1}{N}$ denotes the angle parameter of the *i*-th polar-domain codeword.

In the polar-domain codebook, each codeword is defined by the coupling of angle and distance parameters. For the purpose of adapting to near-field channel characteristics, multiple distance sampling points are associated with each angular direction [33]. For any given angle $\theta_{i}\in \bar{\Theta }$, the corresponding distance sampling values can be determined as follows $r_{i,z}=\frac{1}{z}Z_{\Delta }\left(1-\theta_{i}\right),z=1,\cdots ,Z,$ where $Z_{\Delta }=\frac{N^{2}\lambda }{8\beta_{\Delta }^{2}}$ serves as the coherence threshold for the response vectors of near-field, *Z* denotes the count of distance samples and the parameter $\beta_{\Delta }$ represents the coherence threshold between the neighboring polar-domain codewords, used to partition codewords into polar domains. A larger β indicates lower coherence among codewords within the polar codebook, as shown in Figure 4, where |G(β)| denotes the codeword coherence. Denote the polar-domain codebook as ${\bar{D}}^{n}$, containing a total of $\bar{M}\times Z$ codewords, where $\bar{M}\;=\;\left|\bar{\Theta }\right|$. Each codeword $b\left(\theta_{i},r_{i,z}\right)\;,i\in A_{t}^{f},z=1,\cdots ,Z$ is defined by a specific angle-distance pair $\left(\theta_{i},r_{i,z}\right)$.

This layer aims to identify an optimal subset of codewords from ${\bar{D}}^{n}$, such that the total beam gains of all scheduled users are maximized, and the problem is written as(14)P3:maxAtn∑t=1T∑k∈Gt∑i∈AtnhtkHdin(14a)s.t.din∈D¯n.

To reduce training overhead, we set $M^{n}=\left|A_{t}^{n}\right|$, where $A_{t}^{n}$ is solved in a manner analogous to that used in the first layer. The two-phase MAB-based beam training procedure is detailed in Algorithm 2.

Adopting an approach analogous to Theorem 1’s proof, this work investigates the exploration–exploitation dilemma inherent to the polar-domain codeword selection process at the second layer. It can be shown that the expected regret in this phase also exhibits an upper bound of O(lnt).

### 3.4. Design of Multi-User Digital Precoding

In the second layer of the second phase, the BS transmits a pilot matrix $X^{H}\in C^{\left|A_{t}^{n}\right|\times \left|A_{t}^{n}\right|}$, and the signal detected at user *k* ($k\in G_{t}$) is formulated by(15)$${\left(y_{k,t}^{n}\right)}^{H}={\left(h_{t}^{k}\right)}^{H}D_{A_{t}^{n}}X^{H}+n_{k,t}.$$To estimate ${\left({\bar{h}}_{t}^{k}\right)}^{H}={\left(h_{t}^{k}\right)}^{H}D_{A_{t}^{n}}$, the pilot matrix must satisfy orthogonality condition $X^{H}X=I$. Under the premise that the condition is satisfied, the Least Squares channel estimation is employed. Then, we have(16)$${\left(y_{k,t}^{n}\right)}^{H}X={\left(h_{t}^{k}\right)}^{H}D_{A_{t}^{n}}+n_{k,t}X,$$
where the estimated value of ${\left({\bar{h}}_{t}^{k}\right)}^{H}$ is ${\left({\hat{\bar{h}}}_{t}^{k}\right)}^{H}={\left(y_{k,t}^{n}\right)}^{H}X$, and this estimate is fed back to the BS.

Following a procedure analogous to Remark 2, we select a set of $M^{n}$ polar-domain codewords. Among these codewords, the one with the maximum KL-UCB value for user *k* ($k\in G_{t}$) in the current coherence block must be included. We use these codewords to form $A_{t}^{1}$. The BS utilizes $A_{t}^{1}$ and ${\left({\hat{\bar{h}}}_{t}^{k}\right)}^{H}$ to obtain an estimate of ${\bar{H}}_{G_{t}}^{H}=H_{G_{t}}^{H}D_{A_{t}^{1}}$, denoted as ${\hat{\bar{H}}}_{G_{t}}^{H}$. To mitigate interference, the multi-user digital precoder $V_{t}$ takes the form of ${\left({\hat{\bar{H}}}_{G_{t}}^{H}\right)}^{†}$. While ideal zero-forcing requires perfect CSI, our approach follows practical implementations where estimated CSI is employed for precoder design [23,27,43]. This represents a trade-off between performance and practicality, effectively suppressing interference when channel estimation quality is sufficiently high.

## 4. Complexity Analysis

The computational complexity of the proposed two-stage C-MAB framework originates from three parts: energy-aware user scheduling, angular scanning, and polar-domain scanning.

In the user scheduling part, in each coherent block, Algorithm 1 calculates the EA-UCB metric for *K* users. The per-user computation, involving historical average rate, exploration term, and a penalty term, has constant complexity O(1). Selecting the top-*M* users requires sorting, with complexity O(K·M). Over *T* coherent blocks, the total complexity is O(T·K·(M+1)).

In the angular scanning part, Algorithm 2 employs a DFT codebook of size *N*. KL-UCB indices for *N* codewords are computed via a bisection method. The KL divergence for the exponential reward distribution is solved with the complexity $O\left(1\right)$, and the bisection requires $C_{a}$ iterations. Selecting $M^{f}$ candidate beams has complexity $O\left(N\cdot M^{f}\right)$. The per-block complexity is, therefore, $O\left(N\cdot (C_{a}+M^{f})\right)$, leading to an overall complexity of $O\left(T\cdot N\cdot (C_{a}+M^{f})\right)$.

The polar-domain scanning part searches over a candidate set with $M^{f}$ angles, each associated with *Z* distance samples. The candidate codebook size is $O\left(M^{f}\cdot Z\right)$. Using the same KL-UCB logic as polar-domain scanning to select $M^{n}$ candidate beams, the bisection requires $C_{p}$ iterations, so its per-block complexity is $O\left(M^{f}\cdot Z\cdot C_{p}\right)$ and overall complexity is $O\left(T\cdot M^{f}\cdot Z\cdot \left(1+C_{p}+M^{n}\right)\right)$.

Integrating the three parts, the total time complexity of the proposed framework is $O\left(T\cdot [K\cdot \left(M+1\right)+N\cdot \left(C_{a}+M^{f}\right)+M^{f}\cdot Z\cdot \left(1+C_{p}+M^{n}\right)]\right)$.

The graph-based I-CSI algorithm (ICSIG) [13] requires full instantaneous CSI acquisition and solves an NP-hard maximum-weight clique problem, resulting in a complexity of $O\left(T\cdot [K^{2}+1.1996^{K}+M\left(N\cdot Z\right)]\right)$ [44]. The Thompson-sampling-based beam training (TPBT) [45,46] has a complexity of $O\left(K+K\cdot M\right)$ in the user schedule part, whille has a complexity of $O\left(\log(NZ)\right)$ in the two-stage beam training part. So the total time complexity of the TPBT is $O\left(T\cdot [K+K\cdot M+\log(NZ)]\right)$.

## 5. Numerical Analysis

This section presents a performance evaluation of the introduced energy-aware US-NFBT scheme. The simulation parameters and channel configurations are summarized in Table 1. Let $r_{k,l}$ be the distance between the *l*-th cluster of user *k* and the BS. The parameter $r_{k}^{\prime }$ is normalized to resolve magnitude discrepancies in the EA-UCB formulation, and $r_{k}^{\prime }=\frac{\mathrm{ave}\left(r_{k,l}\right)-\min\left(r_{k,l}\right)}{\max\left(r_{k,l}\right)-\min\left(r_{k,l}\right)}$, where $\mathrm{ave}\left(r_{k,l}\right)$, $\min(r_{k,l})$ and $\max(r_{k,l})$ are the average, minimum, and maximum distances between the clusters of user *k* and the BS, respectively. This normalization prevents any single component from dominating the scheduling decision and ensures balanced consideration of both distance and energy factors in the EA-UCB algorithm.

Figure 5 illustrates the performance of the proposed scheme under the scenario where user energy is enough. The system can consistently schedule users with the best channel conditions, allowing the EAR to remain at a high level after convergence. The proposed hierarchical MAB framework has a theoretical logarithmic regret upper bound O(lnt) (**Theorem 1**), meaning it converges to a near-optimal policy within 20–30 coherence blocks. This is significantly shorter than the 250-coherence-block S-CSI invariance interval, ensuring the algorithm completes learning before channel statistics change. Cumulative regret arises primarily from the initial exploration phase and grows slowly due to the KL-UCB and EA-UCB strategies.

The proposed budget-based MAB scheme for is compared with two baseline schemes:ICSIG [13]: This scheme is adapted from [13] and relies on I-CSI. It first obtains I-CSI through channel estimation, and then employs a clique-based approach from graph theory for user scheduling.TPBT [45,46]: This scheme uses classical Thompson sampling [45] for user scheduling, combined with a two-stage beam training strategy [46]. Finally, the estimated S-CSI is utilized to design the digital precoding.

Two scattering scenarios are considered: single cluster (S=1) and triple clusters (S=3).

Figure 6 compares the EAR of all schemes with different signal-to-noise ratios (SNRs). A positive correlation between SNR and EAR is evident for all schemes, indicating a consistent performance gain, which is consistent with theoretical expectations. The US-NFBT scheme consistently outperforms the baselines, owing to its low pilot overhead and efficient user scheduling mechanism, which adaptively selects users with good channel conditions and sufficient energy. In contrast, TPBT considers only channel quality and ignores energy constraints, while ICSIG may miss optimal users due to channel correlation and energy limitations. A more complex scattering environment (S=3) offers greater channel diversity. The rate reduction in the multi-cluster scenario can be attributed to the power normalization in the channel model, thereby lowering the EAR. At high SNR, performance is predominantly governed by channel gain, allowing the TPBT to be temporarily effective. The subsequent performance convergence indicates that the slight short-term penalty introduced by our energy-aware constraint is diminished in this regime, highlighting its advantage for long-term sustainability.

Figure 7 illustrates the EAR of all schemes with different *M*. As shown in Figure 7, the US-NFBT scheme maintains the highest EAR as the number of scheduled users grows. This advantage stems from its ability to dynamically balance channel quality and residual energy. When the best user depletes its energy, US-NFBT adaptively switches to energy-sufficient sub-optimal users, thereby sustaining system performance. Conversely, TPBT suffers from energy exhaustion, and ICSIG may form sub-optimal schedules due to its structural and energy constraints. This adaptability enables US-NFBT to perform well under different numbers of scheduled users. Moreover, these performance advantages are also evident in the single-cluster scenario with varying SNR levels, as shown in Table 2 where US-NFBT consistently outperforms other schemes across all SNR values.

As illustrated in Figure 8, the EAR performance of all schemes under different *K* is presented. US-NFBT maintains the highest EAR as *K* increases, demonstrating superior scalability and stability. This advantage stems from its MAB-based learning framework, which efficiently identifies and prioritizes users with favorable channel conditions and sufficient energy reserves. Consequently, US-NFBT avoids the performance degradation inherent to ICSIG, which suffers from high computational complexity and inflexible rules. It also overcomes the short-sightedness of TPBT in energy-limited scenarios, making it a more suitable solution for large-scale IoT applications.

To provide a more comprehensive performance evaluation, the impact of computational processing delay on the EAR is analyzed under a configuration with SNR = 20 dB, S=1 scattering cluster, K=9 users, and M=5 maximum schedulable users. As shown in Figure 9, the system EAR exhibits a clear decreasing trend as the computational delay increases from 0 ms to 1 ms. This performance degradation is governed by a modified EAR formulation $E_{k,t}^{\mathrm{delay}}=G_{k,t}\left(\frac{T_{\mathrm{total}}-T_{B,t}-T_{\mathrm{delay},t}}{T_{\mathrm{total}}}\right)\log_{2}\left(1+\frac{P_{\mathrm{signal}}}{P_{\mathrm{interference}}+\sigma^{2}}\right)$, which, compared to the original expression in Equation (Equation 7), incorporates an additional overhead term $T_{\mathrm{delay},t}$ to account for the additional overhead introduced by algorithmic processing [47], thereby directly reducing the effective symbols available for data transmission within a coherence block. According to Figure 9, due to the inherent exploration–exploitation dynamic mechanism of the multi-arm slot machine framework adjusting for the delayed changes, some minor fluctuations were observed. The overall trend of the curve is that the performance decreases as the delay increases. Importantly, the proposed US-NFBT scheme maintains robust performance under moderate delays, validating its practical applicability in real-world scenarios where processing delays is present [47].

## 6. Conclusions

This paper presents a C-MAB-driven, two-phase joint optimization framework that jointly designs the beam training and user scheduling in XL-MIMO systems, thereby bridging a significant research gap in FDD near-field communication mechanisms. By incorporating an EA-UCB user scheduling algorithm, terminal energy constraints are integrated into the scheduling decision process, enabling joint optimization of the communication performance and energy consumption. Subsequently, a dual-layer beam training mechanism with coarse-grained angle estimation and advances to fine-grained polar-domain beam training scheme is introduced to substantially cut the training overhead. Simulation results confirm its superior performance. In the future, we will extend our proposed scheme to the non-stationary scenarios, such as scenarios involving high-speed mobile users or time-varying scatterers. By incorporating adaptive learning mechanisms and dynamic model prediction, we will enhance the system’s robustness and adaptability in these scenarios. 

## Figures and Tables

**Figure 1 sensors-26-00060-f001:**
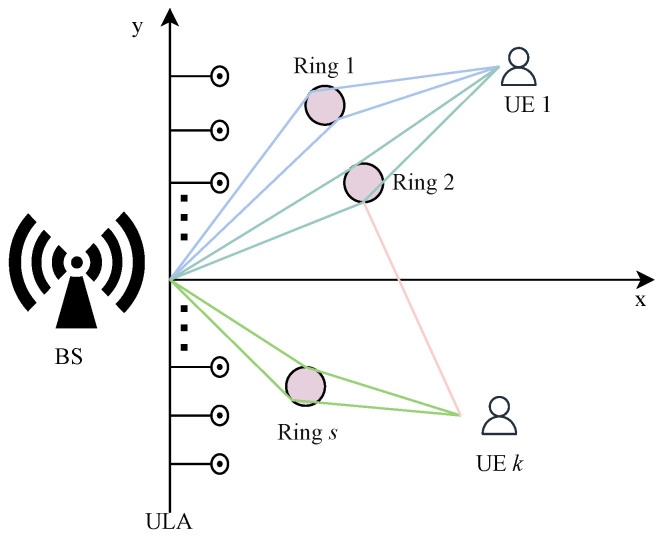
Illustration of the XL-MIMO multi-user communication system.

**Figure 2 sensors-26-00060-f002:**
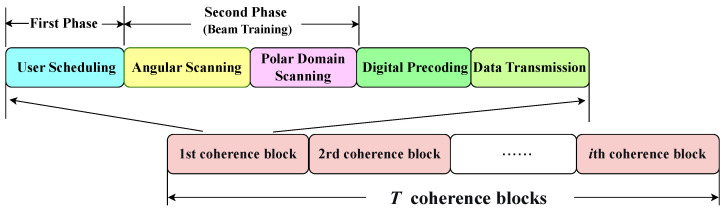
The US-NFBT Scheme Architecture.

**Figure 3 sensors-26-00060-f003:**
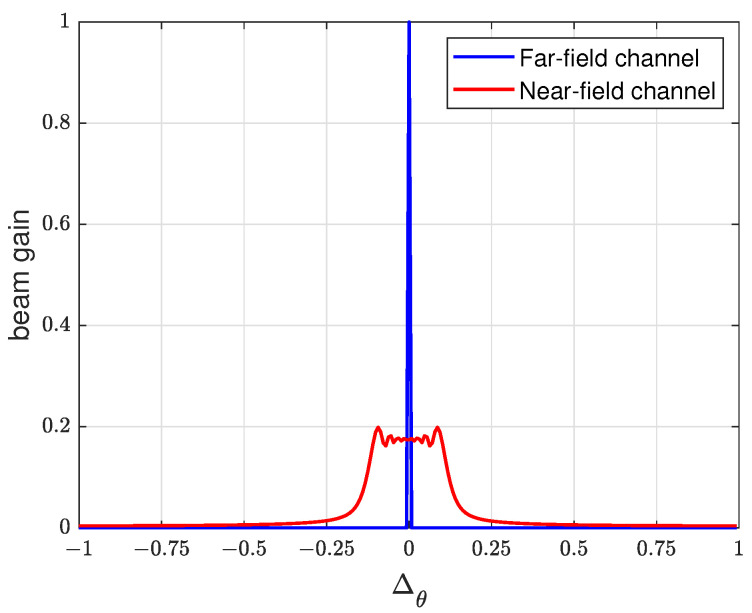
Comparison between the far-field and near-field channel.

**Figure 4 sensors-26-00060-f004:**
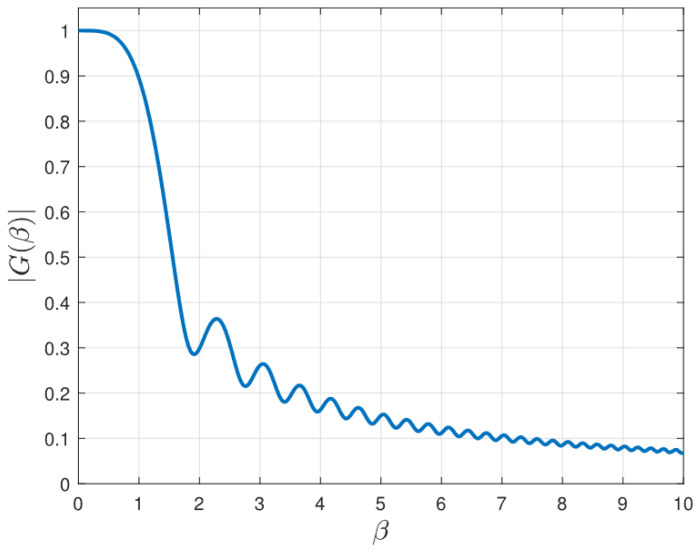
The numerical results of |G(β)| against β [42].

**Figure 5 sensors-26-00060-f005:**
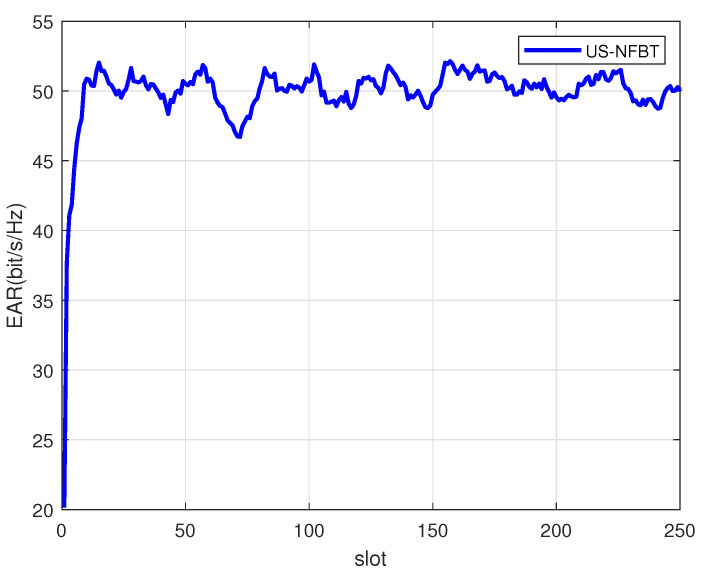
Comparison of EARs under different slots. SNR = 20 dB, S=1, K=9, M=5.

**Figure 6 sensors-26-00060-f006:**
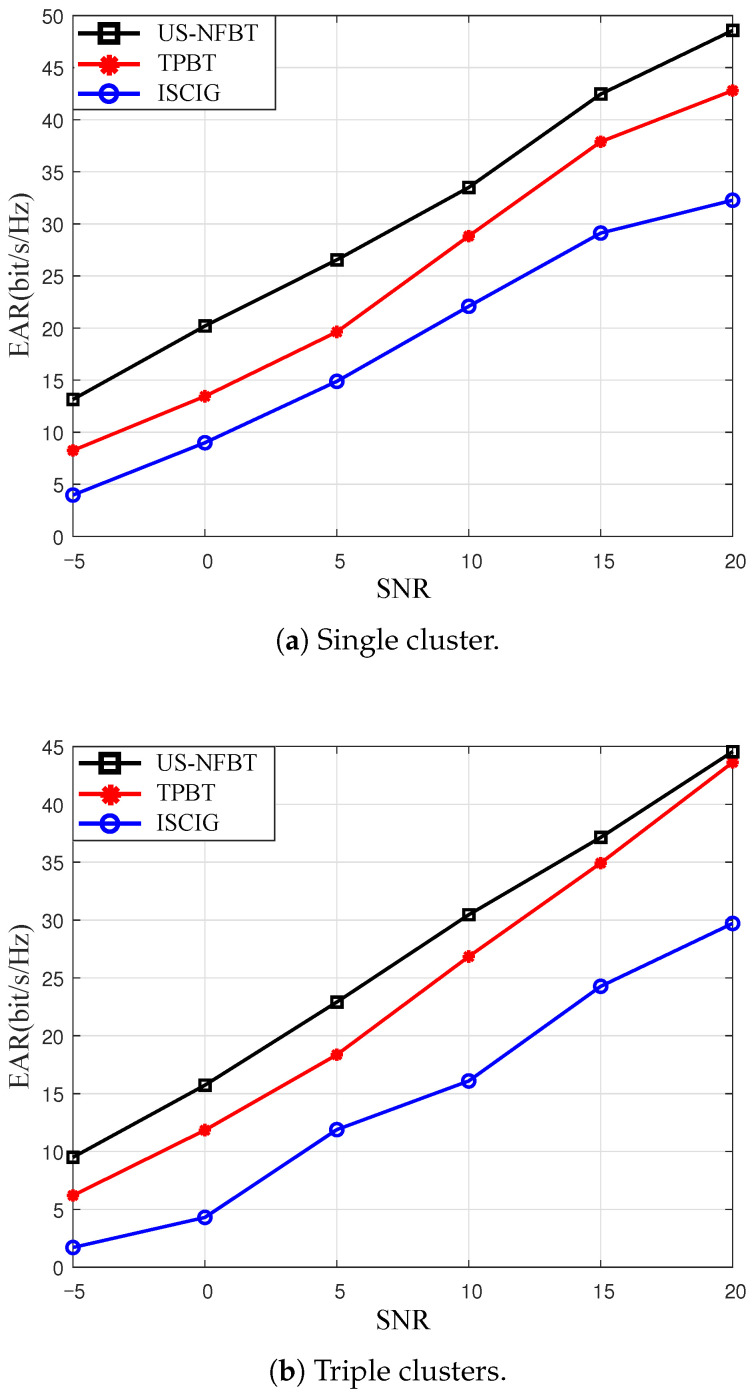
The total EARs versus different SNRs for US-NFBT, TPBT [45,46], ICSIG [13] schemes. K=12, M=5.

**Figure 7 sensors-26-00060-f007:**
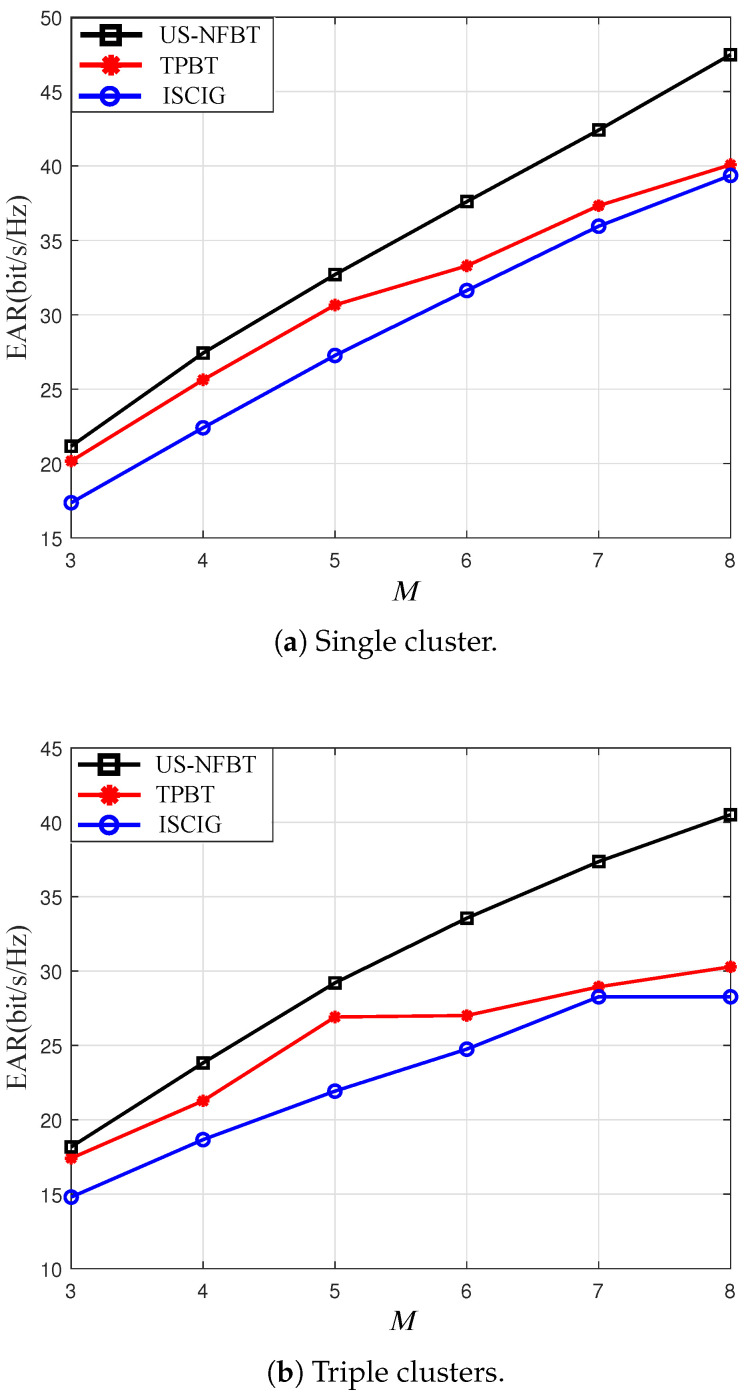
Total EARs versus *M* for US-NFBT, TPBT [45,46], ICSIG [13] schemes. SNR = 10 dB, K=12.

**Figure 8 sensors-26-00060-f008:**
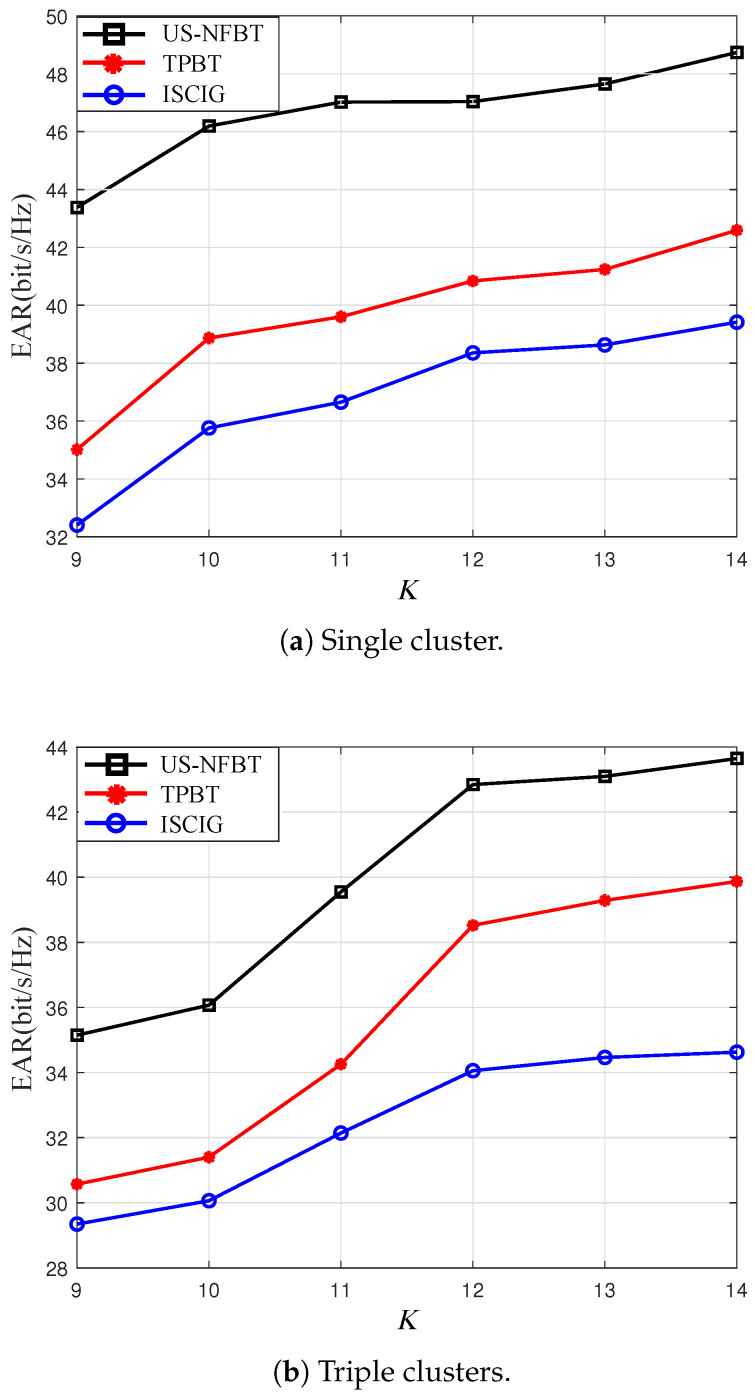
The total EARs versus *K* for US-NFBT, TPBT [45,46], ICSIG [13] schemes. SNR = 20 dB, M=5.

**Figure 9 sensors-26-00060-f009:**
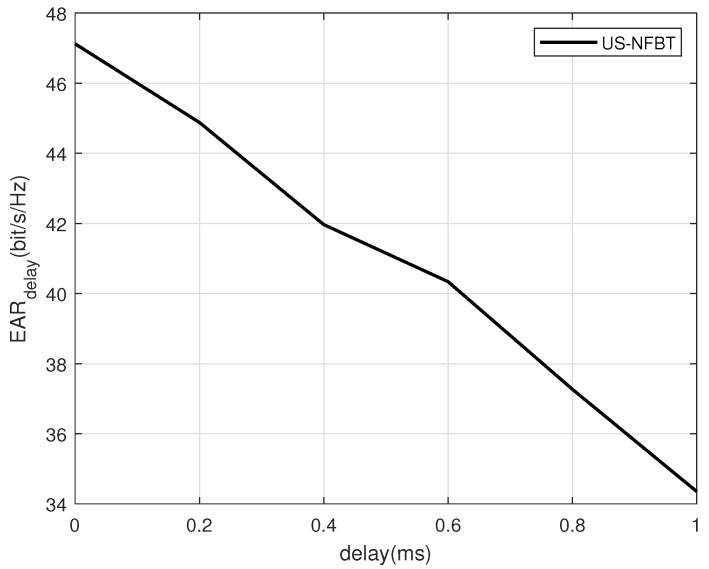
Comparison of ${\mathrm{EAR}}_{\mathrm{delay}}$ under different computational delay.

**Table 1 sensors-26-00060-t001:** Simulation Parameters.

Parameter	Value
**System**	
Antenna count *N*	256
Frequency of carrier $f_{c}$	100 GHz
Antennas spacing *d*	1.5 mm
System circuit power $\xi_{\mathrm{system}}$	44 dBm
Efficiency of the power amplifier 1/γ	1/3
Predefined energy threshold $\Psi_{\mathrm{th}}$	10 Joule
Bandwidth $W_{\mathrm{bw}}$	63 MHz
Initial user battery energy $\Psi_{k,1}$	30 Joule
Number of symbols per coherence block $T_{\mathrm{total}}$	1386
User number *K*	$\left[9,14\right]$
Required data per coherence block $L_{\mathrm{data}}$	1.5 Mb
Maximum number of scheduled users *M*	$\left[3,8\right]$
**Channel**	
User angle range $\theta_{k}$	$\left\{\pm 60^{\circ },\pm 45^{\circ },\pm 30^{\circ },\pm 15^{\circ },0^{\circ }\right\}$
Number of clusters *S*	$\left\{1,3\right\}$
Scattering cluster distance range $r_{k,l}$	$\left\{6,8,10,12,14\right\}$
S-CSI invariance interval	250 coherence block
Concentration parameter of the von-Mises PDF	$\left\{0,4,8\right\}$
**MAB**	
Exploration coefficient of EA-UCB *B*	$\log_{2}\left(1+\frac{\max_{k}\left|{\left(d_{i}^{n}\right)}^{H}R_{k}d_{i}^{n}\right|}{\sigma^{2}}\right)$
Tuning coefficient of KL-UCB *c*	3
Coefficient ρ (**Remark 2**)	0.5

**Table 2 sensors-26-00060-t002:** EAR for different schemes at various SNR in single-cluster scenario.

SNR (dB)	EAR (bit/s/Hz)		
	**US-NFBT**	**TPBT**	**ISCIG**
−5	13.17	8.93	4.14
0	20.19	13.63	8.86
5	26.54	19.93	15.04
10	33.50	28.53	22.42
15	42.45	37.56	28.86
20	48.57	43.21	32.47

## Data Availability

Dataset available on request from the authors.

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
