# Peer review of "Hierarchical MAB Framework for Energy-Aware Beam Training for Near-Field Communications"

_sensors, 2025, doi:10.3390/s26010060_

Round 1
Reviewer 1 Report
Comments and Suggestions for Authors
The overall presentation of the submission is quite satisfactory. Although, there are some critical issues which prevent me from recommending the submission for publication in its current form.
1.Please, clarify the mathematical model.
– First, please cite the number of the equation in [19] that corresponds to (1). It seems that the closest ones are (3)-(4) or (33)-(34), but the normalization factors are very different. Moreover, it seems more natural that the denominator should account for the total number of paths across all clusters and all users if power normalization is intended system-wide, or it should be clarified whether normalization is per-user or global.
– In equation (2), the distances are calculated incorrectly. There should be a cosine, not a sine term (see equation (2) in [19]), or the geometry needs clarification.
– In line 332, please clarify what βΔ is. Up to now, it seems that there is a problem with dimensionality.
2. There are some problems with the assumptions that can be at least weak or even wrong. For example:
– Lines 256-260 claim a DFT codebook Df is used for angular scanning in the near-field. However, DFT codebooks are inherently designed for far-field plane wave assumptions and provide suboptimal angular resolution in near-field scenarios with spherical wavefronts.
– The authors assume a zero-forcing criterion for digital precoding, but zero-forcing requires perfect CSI. As far as I see, the paper uses estimated CSI with estimation error, making true zero-forcing impossible.
3. In the simulation part, it is assumed that the cluster distances are normalized. But how do the authors introduce such a normalization? And what is it intended for?
4.There are some flaws that seem to lead to physically meaningless conclusions. For example, equation (6) models energy consumption as inversely proportional to the achievable rate, which is physically questionable. Energy should be consumed based on transmission power and time, not achievable rate. A user with a poor channel (low rate) consuming more energy contradicts the model in Equation (4). This requires fundamental clarification.
5. The submission lacks analysis of convergence time. For example, how many coherence blocks are there until the MAB algorithms converge? What is the regret accumulated before convergence? For the 250-block S-CSI interval, does convergence occur before the channel statistics change? These are critical for practical deployment. Moreover, complexity analysis is almost absent and thus must be provided.
Author Response
Please see the attached response letter for detailed responses.

Reviewer 2 Report
Comments and Suggestions for Authors
The manuscript introduces a two–phase combinatorial MAB framework that combines energy–aware user scheduling with hierarchical near–field beam training for XL–MIMO systems. The topic is timely and the approach is interesting, especially given the growing attention to energy-efficiency in large antenna-array systems. Howver, before recommending acceptance I have some major concerns that I think the authors should address more clearly.
- The part that links the residual battery level to the scheduling decisions is central to the whole framework, but at the moment it feels a bit “black-boxed”. It would help if the authors could clarify how realistic their energy model is supposed to be, and which assumptions hold across different types of terminals.
- The paper claims that the proposed method reduces the complexity, however there isn’t much quantitative insight into how the algorithm scales when the XL-MIMO size grows (more antennas, more users, maybe more beams to train). Some rough complexity analysis or even a simple scaling experiment would be useful.
- The evaluation focuses mainly on Thompson sampling and a graph-based scheduler. These are reasonable starting points, but the field is moving fast and several more recent scheduling/beam-training strategies could be considered, or at least briefly discussed.
- In the results section the authors report EAR trends under different SNR values. While the absolute performance is good, I would really like to see a metric that accounts jointly for EAR and computational time (or delay). This would give a more complete and fair comparison with the baselines, especially since the paper emphasizes reduced overhead. I suggest adding a latency-weighted performance metric, similar to the one described in [Ref1], Sec. VI-D, so that both efficiency and timing are reflected together.
[Ref1] Miuccio et al., "A Flexible Encoding/Decoding Procedure for 6G SCMA Wireless Networks via Adversarial Machine Learning Techniques," in IEEE Transactions on Vehicular Technology, vol. 72, no. 3, pp. 3288-3303, March 2023, doi: 10.1109/TVT.2022.3216028
Author Response

(The authors gave the same response as above.)

Reviewer 3 Report
Comments and Suggestions for Authors
The article entitled "Hierarchical MAB Framework for Energy-Aware Beam Training for Near-Field communications" presented. MAB framework was employed for near field beam training for extra large MIMO FDD systems. UCB algorithm has been designed to evaluate the performance of user battery levels. numerical results are presented. I have following suggestions to further improve the quality or the article.
- in the abstract, pls mention the DFT acronym.
- pls add 2 more most relevant keywords.
- improve the quality of Figure 2, it is hard to read in the current form.
- pls double check the equations, variables and symbols carefully.
- provide the performance compare table with the recent quality article.
- what are the future directions of the proposed work, pls include it in the Conclusions Section.
Author Response

(The authors gave the same response as above.)

Round 2
Reviewer 1 Report
Comments and Suggestions for Authors
The issues raised by me were carefully addressed.
Reviewer 2 Report
Comments and Suggestions for Authors
The reviewers addressed all my concerns.